# Progression of coronary artery calcification in conventional hemodialysis, nocturnal hemodialysis, and kidney transplantation

Thijs T. Jansz[1,2], Akin Özyilmaz[3,4], Franka E. van Reekum[1], Franciscus T. J. Boereboom[2], Pim A. de Jong[5], Marianne C. Verhaar[1], Brigit C. van Jaarsveld[6]*

1 Department of Nephrology and Hypertension, University Medical Center Utrecht, Utrecht University, Utrecht, The Netherlands, 2 Dianet Dialysis Centers, Utrecht, The Netherlands, 3 Dialysis Center Groningen, Groningen, The Netherlands, 4 Division of Nephrology, Department of Internal Medicine, University Medical Center Groningen, Groningen, The Netherlands, 5 Department of Radiology, University Medical Center Utrecht, Utrecht, the Netherlands, 6 Department of Nephrology and Amsterdam Cardiovascular Sciences (ACS), Amsterdam University Medical Center, Vrije Universiteit Amsterdam, Amsterdam, the Netherlands

* b.jaarsveld@amsterdamumc.nl

## Abstract

### Introduction

Cardiovascular disease is the leading cause of death in end-stage renal disease (ESRD) and is strongly associated with vascular calcification. An important driver of vascular calcification is high phosphate levels, but these become lower when patients initiate nocturnal hemodialysis or receive a kidney transplant. However, it is unknown whether nocturnal hemodialysis or kidney transplantation mitigate vascular calcification. Therefore, we compared progression of coronary artery calcification (CAC) between patients treated with conventional hemodialysis, nocturnal hemodialysis, and kidney transplant recipients.

### Methods

We measured CAC annually up to 3 years in 114 patients with ESRD that were transplantation candidates: 32 that continued conventional hemodialysis, 34 that initiated nocturnal hemodialysis (≥4x 8 hours/week), and 48 that received a kidney transplant. We compared CAC progression between groups as the difference in square root transformed volume scores per year (ΔCAC SQRV) using linear mixed models. Reference category was conventional hemodialysis.

### Results

The mean age of the study population was 53 ±13 years, 75 (66%) were male, and median dialysis duration was 28 (IQR 12–56) months. Median CAC score at enrollment was 171 (IQR 10–647), which did not differ significantly between treatment groups (P = 0.83). Compared to conventional hemodialysis, CAC progression was non-significantly different in nocturnal hemodialysis -0.10 (95% CI -0.77 to 0.57) and kidney transplantation -0.33 (95% CI -0.96 to 0.29) in adjusted models.

**Data Availability Statement:** The data are contained in the Supporting information files (S1 Dataset).

**Funding:** The NOCTx study was supported by unrestricted grants from Amgen, Baxter, Fresenius Medical Care, Novartis, Roche and Shire Pharmaceuticals. T.T. Jansz was supported financially by a grant from the Wellerdieck de Goede Foundation with mediation from Friends of UMC Utrecht. The funders had no role in study design, data collection and analysis, decision to publish, or preparation of the manuscript.

**Competing interests:** The NOCTx study is supported by unrestricted grants from Amgen, Baxter, Fresenius Medical Care, Novartis, Roche, and Shire Pharmaceuticals. B.C. van Jaarsveld reports research grants from Fresenius Medical Care, Baxter, Vifor Fresenius Medical Care Renal Pharma, and Nipro outside the submitted work. There are no patents, products in development or marketed products to declare. This does not alter our adherence to all the PLOS ONE policies on sharing data and materials, as detailed online in the guide for authors.

## Conclusions

Nocturnal hemodialysis and kidney transplantation are not associated with significantly less CAC progression compared to conventional hemodialysis during up to 3 years follow-up. Further studies are needed to confirm these findings, to determine which type of calcification is measured with CAC in end-stage renal disease, and whether that reflects cardiovascular risk.

## Introduction

Cardiovascular disease is the leading cause of death among patients with end-stage renal disease [1, 2]. This high cardiovascular mortality is strongly associated with vascular calcification [3, 4], which occurs frequently and progresses rapidly in end-stage renal disease [2, 5]. Vascular calcification can be measured at various sites, such as the coronary arteries, and vascular calcification in end-stage renal disease is promoted by phosphate, which is frequently elevated in end-stage renal disease [6, 7].

Phosphate levels are considerably lower in patients who dialyze longer and more frequently, such as in frequent nocturnal hemodialysis [8, 9]. By improving phosphate control, nocturnal hemodialysis could mitigate progression of vascular calcification, but progression of vascular calcification in nocturnal hemodialysis has never been compared to conventional hemodialysis. Similarly, it is thought that kidney transplantation could halt progression of vascular calcification. However, only two previous studies compared progression of vascular calcification between kidney transplant recipients and patients on hemodialysis [10, 11]. These studies had important limitations, as in one study those on hemodialysis were not transplantation-eligible and therefore not comparable to kidney transplant recipients [11], whereas both studies did not account for calcification at baseline, which is strongly associated with progression [12].

We therefore set out to compare progression of vascular calcification between conventional hemodialysis, nocturnal hemodialysis, and kidney transplantation. To this end, we conducted the NOCTx study, a prospective study that measured coronary artery calcification (CAC) annually during 3 years in 3 groups of patients with end-stage renal disease: transplantation-eligible patients that were treated with conventional hemodialysis, transplantation-eligible patients that switched from conventional to nocturnal hemodialysis, and patients on dialysis that received a kidney transplant.

## Materials and methods

### Study design and population

NOCTx (NCT00950573) is a prospective study designed to compare CAC progression between different renal replacement therapies. NOCTx included patients that continued chronic conventional hemodialysis or peritoneal dialysis after at least 2 months on dialysis, patients that switched from conventional hemodialysis to nocturnal hemodialysis (≥4x 8 hours per week), and patients on dialysis who received a kidney transplant 2–3 months before enrollment. Patients were eligible when aged between 18 and 75 years and were candidates for transplantation when on dialysis. NOCTx excluded patients with a life expectancy <3 months, pre-emptive transplantation, non-adherence to dialysis regimens, drug abuse, and pregnancy. All participants gave written informed consent. NOCTx has been approved by the Medical

Ethics Committee of the University Medical Center Utrecht and was conducted according to the Declaration of Helsinki.

Between December 2009 and February 2016, 329 patients were screened for eligibility in 8 Dutch dialysis centers including two academic hospitals where transplantation procedures took place during that period. NOCTx included 181 of these patients, who underwent study exams at University Medical Center Utrecht at enrollment and after 1, 2, and 3 years. Patients left the study if they switched renal replacement therapy, except for 8 patients on dialysis that received a kidney transplant within 6 months after enrollment and continued participation in the kidney transplantation group. For the current study, we excluded patients treated with peritoneal dialysis (*n = 31*).

### Treatment characteristics

Patients were treated according to the Kidney Disease: Improving Global Outcomes (KDIGO) 2009 guidelines by the attending nephrologists [13]. Conventional hemodialysis (3x 4 hours per week, in-center) and nocturnal hemodialysis (4-6x 8 hours per week, at home) were delivered with a default dialysate calcium concentration of 1.50 mmol/L. Kidney transplant recipients received standard immunosuppressant regimens consisting of a calcineurin inhibitor (tacrolimus), mycophenolate mofetil, and prednisolone in tapering doses. All live donors provided written informed consent that was freely given. Deceased donors were nationally registered organ donors whose organs were allocated by Eurotransplant (Leiden, the Netherlands).

### CAC measurements

CAC scores were determined at each study exam using non-enhanced prospectively triggered cardiac multi-slice computed tomography (iCT 256 or IQon, Philips Medical Systems, Best, the Netherlands). Acquisition parameters were as follows: 120 kV, 40–50 mAs, rotation time 270 ms, and 128 x 0.625 mm collimation (iCT 256) / 64 x 0.625 mm collimation (IQon). To improve imaging quality, metoprolol was given intravenously if heart rate was above 60/min. We used a calcium threshold of $\geq$130 Hounsfield units. A single reader (TJ) read all scans blinded for treatment group and chronologically per patient, in order to exclude coronary segments with severe motion artefacts or stents at a given scan from an entire set. We used calcium volume scores as primary outcome measure and also calculated Agatston scores. These scores are highly correlated (Spearman's $\rho$ = 0.99). Reproducibility of coronary artery calcification measurements has been shown to be excellent (intraclass correlation coefficient >0.95) [14].

### Other variables

At each study exam, study personnel collected data on laboratory parameters (total calcium, albumin, phosphate, and parathyroid hormone) by averaging values of routine measurements of the 3 months preceding the study exam. Residual urine production was classified as present ($\geq$100 mL/24h) or absent. We assessed history of kidney disease, current dialysis schedule, and presence of comorbidities by chart review, and evaluated medication use by medication inventory.

We estimated glomerular filtration rate with the Chronic Kidney Disease Epidemiology Collaboration (CKD-EPI) 2009 equation for kidney transplant recipients. We defined dialysis duration as the time between the first day of dialysis and enrollment, minus the time with a functioning kidney transplant. Cardiovascular events were defined as any myocardial infarction, percutaneous coronary intervention, coronary artery bypass grafting, aortic aneurysm repair, stroke, new intermittent claudication, peripheral artery angioplasty or bypass grafting.

## Statistical analyses

We reported normally distributed variables as mean (± standard deviation), non-normally distributed variables as median (interquartile range, IQR), and categorical data as number (percentage). We compared normally distributed variables with Student's t-tests between two groups and with one-way analyses of variance between three, non-normally distributed variables with Mann-Whitney-U tests between two groups and with Kruskal-Wallis tests between three, and categorical variables with chi-squared tests.

The associations of treatment group with CAC progression were evaluated by defining CAC progression as change per year in square root transformed volume scores (ΔCAC SQRV). This approach, also known as Hokanson's method, accounts for interscan variability [15] and has been used by others [16, 17]. The outcome variable in this approach (ΔCAC SQRV) is normally distributed, enabling adjustments for potential confounders. We adjusted these analyses for CAC SQRV at enrollment and used mixed-effects to account for repeated measurements. We adjusted for factors related to calcification [18]: age (years), sex, presence of diabetes mellitus, dialysis duration (months), current smoking, presence of residual urine production, and vitamin K antagonist use. Conventional hemodialysis was the reference group. Of note, the comparison of CAC progression in conventional hemodialysis and peritoneal dialysis is discussed elsewhere [19].

Regression coefficients are reported with 95% confidence intervals (95% CI). We considered p-values of $\leq 0.05$ (two-tailed) statistically significant and used R 3.4.1 (R Foundation Statistical Computing) for all analyses.

## Sensitivity analyses

To test the robustness of the associations, we repeated the analyses of treatment group with CAC progression using Agatston scores instead of volume scores.

## Results

### Study population

A total of 181 patients were included in the NOCTx study. For the current study, we excluded patients treated with peritoneal dialysis (*n = 31*) and patients that did not attend any follow-up exam (*n = 36*), leaving an analytical sample of 114 patients (Fig 1). The mean age of the study population (*n = 114)* was 53 ±13 years, 75 (66%) were male, dialysis duration (including historical dialysis duration of kidney transplant recipients) was median 28 (IQR 12–56) months, and 14 (12%) had diabetes mellitus (Table 1). There were 32 patients treated with conventional hemodialysis, 34 treated with nocturnal hemodialysis, and 48 kidney transplant recipients. Patients on nocturnal hemodialysis were enrolled after a training period of about 3 months. Kidney transplant patients were included about 3 months after transplantation, since many (25/48) received a deceased-donor transplant. Compared to patients on conventional hemodialysis, patients on nocturnal hemodialysis less often used vitamin D analogs (42% versus 84%) or calcium-containing phosphate binders (15% versus 50%). Also, patients on nocturnal hemodialysis had lower phosphate levels than patients on conventional hemodialysis (mean 1.2 versus 1.5 mmol/L), as did kidney transplant recipients (mean 0.9 mmol/L). These differences in phosphate levels subsisted during follow-up (S1 Fig).

Patients that did not complete any follow-up exam (*n = 36*) had similar characteristics compared to the study population (S1 Table): they were on average 52 ±13 years old, 23 (64%) were male, dialysis duration was median 36 (IQR 18–66) months, and 10 (28%) had diabetes

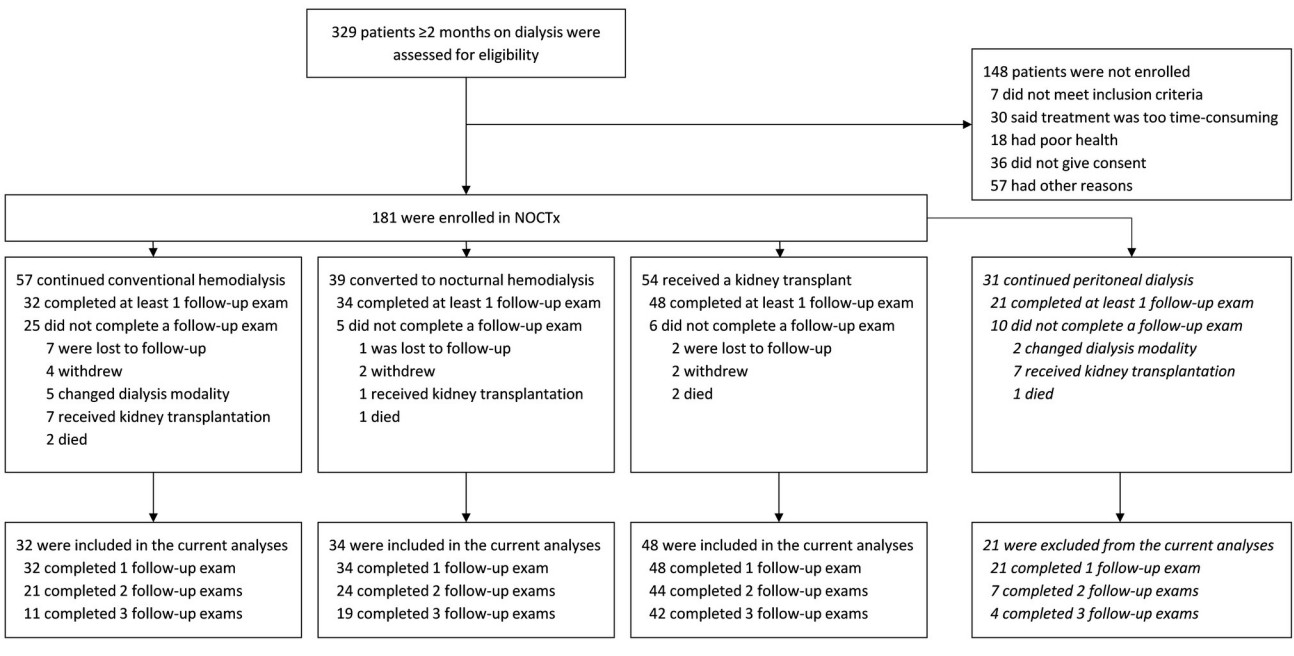

**Fig 1. Study flowchart.**

mellitus (all P>0.05 versus study population). Their median CAC score at enrollment was 323 (IQR 1–1181) (P = 0.18 versus study population).

During follow-up, 2 patients on conventional hemodialysis died (6%), as did 0 patient on nocturnal hemodialysis, and 2 kidney transplant recipients (4%). Cardiovascular events occurred in 2 patients on conventional hemodialysis (6%), in 5 patients on nocturnal hemodialysis (15%), and in 3 kidney transplant recipients (6%). Eleven patients on conventional hemodialysis received a kidney transplant (34%), as did 7 patients on nocturnal hemodialysis (21%).

## Associations of renal replacement therapy with CAC progression

At enrollment, CAC scores were median 182 (IQR 3–835) in patients on conventional hemodialysis, 176 (IQR 16–501) in patients on nocturnal hemodialysis, and 110 (IQR 10–523) in kidney transplant recipients (P = 0.83 for difference). Eight patients on conventional hemodialysis had no calcification (25%), compared to 7 on nocturnal hemodialysis (21%) and 9 kidney transplant recipients (19%).

During 3 years of follow-up, CAC progressed in most patients (Table 2 and Fig 2). In patients on conventional hemodialysis, ΔCAC SQRV was 1.37 per year (95% CI 0.81 to 1.93), while it was 1.29 per year in patients on nocturnal hemodialysis (95% CI 0.77 to 1.82) and 0.89 per year in kidney transplant recipients (95% CI 0.48 to 1.30). Patients on nocturnal hemodialysis and kidney transplant recipients did not have significantly less CAC progression compared to patients on conventional hemodialysis, with confidence intervals overlapping zero in both unadjusted and adjusted analyses (Table 3). CAC progression was also not significantly less in kidney transplant recipients when compared to patients on conventional and nocturnal hemodialysis combined (adjusted difference in ΔCAC SQRV -0.30, 95% CI -0.80 to 0.22). The above associations were similar when we used Agatston scores instead of volume scores (S2 and S3 Tables).

**Table 1. Characteristics at enrollment of 114 patients with end-stage renal disease stratified by renal replacement therapy.**

| | Conventional hemodialysis (n = 32) | Nocturnal hemodialysis (n = 34) | Kidney transplantation (n = 48) | P value |
|---|---|---|---|---|
| *Demographics and medical history* | | | | |
| **Age (years)** | 53 ±12 | 52 ±13 | 52 ±14 | 0.93 |
| **Male sex (%)** | 19 (59%) | 20 (59%) | 36 (75%) | 0.21 |
| **Diabetes mellitus (%)** | 6 (19%) | 5 (15%) | 3 (6%) | 0.22 |
| **Cardiovascular disease (%)** | 7 (22%) | 9 (27%) | 6 (13%) | 0.26 |
| **Current smoker (%)** | 4 (13%) | 6 (18%) | 6 (13%) | 0.77 |
| *History of kidney disease* | | | | |
| **Dialysis duration (months)** | 27 (11–58) | 29 (16–56) | 28 (12–51) | 0.65 |
| **Cause of end-stage renal disease (%)** | | | | 0.28 |
| • Cystic kidney disease | 4 (13%) | 6 (18%) | 14 (29%) | |
| • Interstitial nephritis | 2 (6%) | 0 | 1 (2%) | |
| • Glomerulonephritis | 8 (25%) | 11 (32%) | 9 (19%) | |
| • Vascular disease | 8 (25%) | 4 (12%) | 10 (21%) | |
| • Diabetic nephropathy | 5 (16%) | 2 (6%) | 2 (4%) | |
| • Other | 3 (9%) | 6 (18%) | 5 (10%) | |
| • Unknown | 2 (6%) | 5 (15%) | 7 (15%) | |
| *Dialysis therapy and kidney function* | | | | |
| **Dialysis therapy** | | | | |
| • Weekly dialysis sessions | 2.9 ±0.4 | 5.2 ±0.8 | - | |
| • Weekly dialysis hours | 11.0 ±2.0 | 41.2 ±6.4 | - | |
| **Kidney function** | | | | |
| • Residual urine production ≥100mL/24h (%) | 19 (59%) | 11 (32%) | - | |
| • eGFR (mL/min) | - | - | 57 ±20 | |
| *Medication use* | | | | |
| **Vitamin K antagonists (%)** | 7 (22%) | 5 (15%) | 4 (8%) | 0.23 |
| **Vitamin D analogs (%)** | 27 (84%) | 14 (42%) | 5 (10%) | <0.01 |
| **Calcium-containing phosphate binder (%)** | 16 (50%) | 5 (15%) | - | <0.01 |
| **Cinacalcet (%)** | 7 (22%) | 9 (27%) | 2 (4%) | 0.07 |
| *Physical and laboratory parameters* | | | | |
| **Body mass index (kg/m$^2$)** | 26.8 ±5.0 | 25.8 ±5.4 | 24.4 ±3.5 | 0.06 |
| **Systolic blood pressure (mmHg)** | 142 ±19 | 138 ±19 | 128 ±14 | 0.01 |
| **Diastolic blood pressure (mmHg)** | 79 ±11 | 77 ±11 | 77 ±9 | 0.80 |
| **Calcium (mmol/L)** | 2.3 ±0.1 | 2.3 ±0.2 | 2.4 ±0.1 | 0.02 |
| **Albumin (g/L)** | 40.8 ±3.1 | 41.7 ±3.6 | 39.9 ±3.4 | 0.07 |
| **Phosphate (mmol/L)** | 1.6 ±0.3 | 1.2 ±0.3 | 0.9 ±0.5 | <0.01 |
| **Parathyroid hormone (pmol/L)** | 30 (17–48) | 5 (3–14) | 11 (7–21) | <0.01 |

Data are presented as mean ±standard deviation, median (interquartile range) or number (percentage).

Data of patients on nocturnal hemodialysis and kidney transplant recipients were measured at enrollment, i.e. about 3 months after initiating this treatment.

Abbreviations: eGFR: estimated glomerular filtration rate, calculated with the Chronic Kidney Disease-Epidemiology Collaboration equation 2009.

## Discussion

In this study, we investigated CAC progression measured annually among patients treated with conventional hemodialysis, nocturnal hemodialysis, and kidney transplant recipients, during up to 3 years of follow-up. Our study shows that nocturnal hemodialysis is not associated with less CAC progression compared to conventional hemodialysis. Furthermore, our

**Table 2. Coronary calcium scores at annual follow-up exams in 114 patients with end-stage renal disease.**

|  | *N** | Enrollment | *N* | Year 1 | *N* | Year 2 | *N* | Year 3 |
|---|---|---|---|---|---|---|---|---|
| **Conventional hemodialysis** | *32* | 182 (3–835) | *32* | 199 (27–1045) | *21* | 106 (20–885) | *11* | 334 (67–835) |
| **Nocturnal hemodialysis** | *34* | 176 (16–501) | *34* | 267 (20–583) | *24* | 464 (46–743) | *19* | 511 (107–799) |
| **Kidney transplantation** | *48* | 103 (10–523) | *48* | 154 (35–559) | *44* | 144 (37–648) | *42* | 194 (48–731) |

Coronary calcium scores in mm$^3$ are presented as median (IQR).

*Patients without any follow-up exams were not included in the current analyses.

study shows no significant difference in CAC progression between kidney transplant recipients and patients on conventional hemodialysis.

To our knowledge, this is the first study to compare CAC progression between conventional hemodialysis and nocturnal hemodialysis. Contrary to expected, we did not find less CAC progression in nocturnal hemodialysis or kidney transplantation compared to conventional hemodialysis, despite the fact that phosphate levels are substantially lower among patients on nocturnal hemodialysis [20] and kidney transplant recipients [21]. An interpretation could be that vascular calcification progresses regardless of the type of renal replacement therapy. This contrasts with the prevailing paradigm that kidney transplant recipients have lower cardiovascular morbidity and mortality in part due to mitigating effects of kidney transplantation on vascular calcification [22]. Rather, based on our current findings, one might also say that kidney transplant recipients have lower cardiovascular morbidity and mortality *despite* progressive vascular calcification. This suggests a discrepancy between progression of vascular calcification and hard endpoints.

Previously, others have also pointed out important discrepancies between the effects of several drugs on vascular calcification and mortality [23]. For example, several trials have demonstrated that drugs such as cinacalcet or sevelamer may slow down progression of coronary artery calcification [24, 25] but lack benefit on mortality [25, 26]. These discrepancies suggest that vascular calcification might only be a secondary phenomenon to vascular damage and may not be harmful in itself. We therefore believe future studies should use major adverse

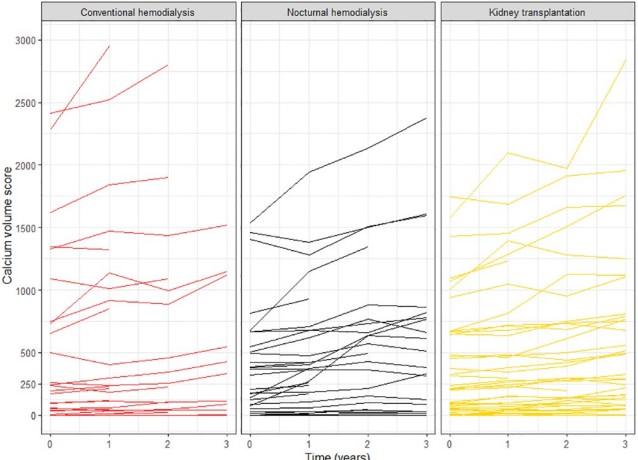

**Fig 2. Progression of coronary artery calcification in 135 patients with end-stage renal disease stratified by renal replacement therapy.** Two patients on conventional hemodialysis are off-scale, with CAC scores at enrollment/after 1 year of 5129/6793 and 5731/7424. Note that lines near zero can overlap.

**Table 3. Longitudinal changes in calcium scores between annual follow-up exams in 114 patients with end-stage renal disease.**

| | N | Mean change per year | Unadjusted difference | Model 1* | Model 2† | Model 3‡ |
|---|---|---|---|---|---|---|
| **Conventional hemodialysis** | 32 | | | | | |
| ΔCAC SQRV | | 1.37 (0.81 to 1.93) | 0.0 (reference) | 0.0 (reference) | 0.0 (reference) | 0.0 (reference) |
| **Nocturnal hemodialysis** | 34 | | | | | |
| ΔCAC SQRV | | 1.29 (0.77 to 1.82) | -0.08 (-0.84 to 0.69) | 0.04 (-0.65 to 0.74) | 0.03 (-0.65 to 0.70) | -0.10 (-0.77 to 0.57) |
| **Kidney transplantation** | 48 | | | | | |
| ΔCAC SQRV | | 0.89 (0.48 to 1.30) | -0.48 (-1.17 to 0.22) | -0.30 (-0.93 to 0.33) | -0.28 (-0.90 to 0.33) | -0.33 (-0.96 to 0.29) |

95% confidence intervals between brackets.

*Model 1 = Adjusted for CAC SQRV at enrollment.

†Model 2 = Model 1 + age and sex.

‡Model 3 = Model 2 + diabetes mellitus, dialysis duration, current smoking, presence of residual urine production, and vitamin K antagonist use.

cardiovascular events or cardiovascular mortality instead of vascular calcification as primary outcome.

On the other hand, our data do not rule out that progression of vascular calcification is less after kidney transplantation compared to dialysis. The 95% confidence intervals of the effect estimates for kidney transplantation include a more than two thirds lower CAC progression rate (adjusted minimum of the 95% confidence interval -0.96), which we consider clinically meaningful. Furthermore, two previous studies reported statistically significant differences in CAC progression with far smaller samples. One study compared CAC progression between 41 kidney transplant recipients and 30 transplantation-eligible patients on hemodialysis during 2 years, and reported less frequent CAC progression among kidney transplant recipients [11]. Another earlier study also reported less frequent CAC progression in 23 kidney transplant recipients compared to 17 patients on hemodialysis during variable follow-up durations, although these patients were not matched on transplantation eligibility [10]. However, regardless of whether CAC progression is less after kidney transplantation, it is unclear to what extent CAC progression concerns progression of medial calcification, which is where an effect of kidney transplantation would be expected. Coronary artery calcification has been suggested to be predominantly intimal [18]. This is based on one post-mortem study in 23 patients on dialysis with known coronary artery disease, which showed that coronary artery calcification was mostly located in intimal plaques, although it also occurred in the media [27]. Moreover, intimal calcification has been suggested to stabilize atherosclerotic plaque [28], which further complicates the interpretation of CAC progression with regard to its potential cardiovascular consequences.

Although our data may be inconclusive regarding the effect of kidney transplantation on CAC progression, they clearly do not suggest less CAC progression in nocturnal hemodialysis compared to conventional hemodialysis despite the lower phosphate levels in nocturnal hemodialysis. The effect estimates for nocturnal hemodialysis were close to zero and do not indicate an undetected difference. Only one previous study reported CAC progression in patients on nocturnal hemodialysis but had no control group [29]. An explanation for similar CAC progression in nocturnal hemodialysis and conventional hemodialysis may be that longer and more frequent hemodialysis could also increase clearance of certain water-soluble calcification inhibitors. These include pyrophosphate [30, 31] and magnesium [32, 33], which have been shown to be lost during hemodialysis. Whether nocturnal hemodialysis indeed results in lower serum levels of pyrophosphate or magnesium needs further study.

This study should be viewed within the context of some limitations. There was significant loss to follow-up, especially among patients on conventional hemodialysis (25 out of 57 enrolled patients). This could have led to survivor bias. Nevertheless, many patients were lost to follow-up due to kidney transplantation (*7/25*) or switch to nocturnal hemodialysis (*5/25*). Therefore, we do not believe that the high rate of loss to follow-up in this group led to a selection of healthier patients. Another limitation is that we could not measure serum levels of certain calcification inhibitors or serum calcification propensity. This could have provided additional information. Furthermore, although this study was the first to compare CAC progression between different dialysis modalities and kidney transplantation with up to 3 years follow-up, sufficiently long to detect CAC progression [34], our sample size was limited and our study was not powered to investigate cardiovascular events or mortality. Finally, our study was observational, which could potentially lead to selection bias. For example, we did not know the reasons for individual patients to opt for nocturnal hemodialysis. On the other hand, we used treatment adherence as an inclusion criterion in order to enroll patients on conventional hemodialysis that were somewhat similar in this respect to patients on nocturnal hemodialysis, who are generally more likely to adhere to treatment. Furthermore, we only enrolled patients on dialysis who were transplantation-eligible and kidney transplant recipients that had been on dialysis before transplantation. Although there may have been unmeasured confounders, patient on dialysis versus kidney transplant recipients were as a result comparable in terms of baseline variables, notably age and dialysis duration.

In conclusion, nocturnal hemodialysis and kidney transplantation are not associated with significantly less CAC progression compared to conventional hemodialysis during up to 3 years follow-up. Further studies are needed to confirm these findings, to determine which type of calcification is measured with CAC in end-stage renal disease, and whether that reflects cardiovascular risk.

## Supporting information

**S1 Table. Characteristics at enrollment of 114 patients who completed at least one follow-up visit and of 36 patients who did not.**
(DOCX)

**S2 Table. Agatston scores at annual follow-up exams in 114 patients with end-stage renal disease.**
(DOCX)

**S3 Table. Longitudinal changes in Agatston scores between annual follow-up exams in 114 patients with end-stage renal disease.**
(DOCX)

**S1 Dataset.**
(CSV)

**S1 Fig.**
(DOCX)

**S1 File.**
(PDF)

**S1 Checklist.**
(DOC)

## Author Contributions

**Conceptualization:** Marianne C. Verhaar, Brigit C. van Jaarsveld.

**Data curation:** Thijs T. Jansz, Brigit C. van Jaarsveld.

**Formal analysis:** Thijs T. Jansz.

**Funding acquisition:** Marianne C. Verhaar, Brigit C. van Jaarsveld.

**Investigation:** Akin Özyilmaz, Franka E. van Reekum, Pim A. de Jong, Marianne C. Verhaar, Brigit C. van Jaarsveld.

**Methodology:** Thijs T. Jansz, Marianne C. Verhaar, Brigit C. van Jaarsveld.

**Project administration:** Franciscus T. J. Boereboom, Marianne C. Verhaar, Brigit C. van Jaarsveld.

**Resources:** Akin Özyilmaz, Franciscus T. J. Boereboom, Pim A. de Jong, Marianne C. Verhaar, Brigit C. van Jaarsveld.

**Software:** Pim A. de Jong.

**Supervision:** Marianne C. Verhaar, Brigit C. van Jaarsveld.

**Visualization:** Thijs T. Jansz.

**Writing – original draft:** Thijs T. Jansz, Brigit C. van Jaarsveld.

**Writing – review & editing:** Thijs T. Jansz, Akin Özyilmaz, Franciscus T. J. Boereboom, Pim A. de Jong, Marianne C. Verhaar, Brigit C. van Jaarsveld.

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
