## [Decision Letter · Decision Letter 0]

31 Jan 2020

PONE-D-19-32023

Progression of coronary artery calcification in conventional hemodialysis, nocturnal hemodialysis, and kidney transplantation

PLOS ONE

Dear Dr. Thijs T Jansz,

Thank you for submitting your manuscript to PLOS ONE. After careful consideration, we feel that it has merit but does not fully meet PLOS ONE’s publication criteria as it currently stands. Therefore, we invite you to submit a revised version of the manuscript that addresses the points raised during the review process.

We would appreciate receiving your revised manuscript by Mar 16 2020 11:59PM. To enhance the reproducibility of your results, we recommend that if applicable you deposit your laboratory protocols in protocols.io, where a protocol can be assigned its own identifier (DOI) such that it can be cited independently in the future. For instructions see: http://journals.plos.org/plosone/s/submission-guidelines#loc-laboratory-protocols

We look forward to receiving your revised manuscript.

Kind regards,

Ping-Hsun Wu, M.D.

Academic Editor

PLOS ONE

2. We note that your study involved tissue/organ transplantation. Please provide the following information regarding tissue/organ donors for transplantation cases analyzed in your study. 1. Please provide the source(s) of the transplanted tissue/organs used in the study, including the institution name and a non-identifying description of the donor(s). 2. Please state in your response letter and ethics statement whether the transplant cases for this study involved any vulnerable populations; for example, tissue/organs from prisoners, subjects with reduced mental capacity due to illness or age, or minors. - If a vulnerable population was used, please describe the population, justify the decision to use tissue/organ donations from this group, and clearly describe what measures were taken in the informed consent procedure to assure protection of the vulnerable group and avoid coercion. - If a vulnerable population was not used, please state in your ethics statement, “None of the transplant donors was from a vulnerable population and all donors or next of kin provided written informed consent that was freely given.” 3. In the Methods, please provide detailed information about the procedure by which informed consent was obtained from organ/tissue donors or their next of kin. In addition, please provide a blank example of the form used to obtain consent from donors, and an English translation if the original is in a different language. 4. Please indicate whether the donors were previously registered as organ donors. If tissues/organs were obtained from deceased donors or cadavers, please provide details as to the donors’ cause(s) of death. 5. Please provide the participant recruitment dates and the period during which transplant procedures were done (as month and year). 6. Please discuss whether medical costs were covered or other cash payments were provided to the family of the donor. If so, please specify the value of this support (in local currency and equivalent to U.S. dollars).

3. We noticed you have some minor occurrence(s) of overlapping text with the following previous publication(s), which needs to be addressed:

https://doi.org/10.1159/000494665

In your revision ensure you cite all your sources (including your own works), and quote or rephrase any duplicated text outside the Methods section. Further consideration is dependent on these concerns being addressed.

4. Thank you for stating the following in the Financial Disclosure section:

"The NOCTx study was supported by unrestricted grants from Amgen, Baxter, Fresenius Medical Care, Novartis, Roche and Shire Pharmaceuticals. T.T. Jansz was supported financially by a grant from the Wellerdieck de Goede Foundation with mediation from Friends of UMC Utrecht. The funders had no role in study design, data collection and analysis, decision to publish, or preparation of the manuscript.".  

We note that you received funding from a commercial source: Amgen, Baxter, Fresenius Medical Care, Novartis, Roche and Shire Pharmaceuticals.

Additional Editor Comments (if provided):

It is important to include baseline serum calcium, serum phosphorus, plasma PTH levels as covariables in multivariate linear regression. A characteristics comparison between enrolled and non-enrolled cases could be considered to reflect the generalizzbility. The lost follow-up is an important issue in this study that the author need to address this issue.

Reviewers' comments:

Reviewer's Responses to Questions

**Comments to the Author**

1. Is the manuscript technically sound, and do the data support the conclusions?

Reviewer #1: Yes

Reviewer #2: Partly

2. Has the statistical analysis been performed appropriately and rigorously? 

Reviewer #1: Yes

Reviewer #2: Yes

3. Have the authors made all data underlying the findings in their manuscript fully available?

Reviewer #1: Yes

Reviewer #2: Yes

4. Is the manuscript presented in an intelligible fashion and written in standard English?

Reviewer #1: No

Reviewer #2: Yes

5. Review Comments to the Author

Reviewer #1: This article entitled “Progression of coronary artery calcification in conventional hemodialysis, nocturnal hemodialysis, and kidney transplantation” by Jansz TT et al investigated progression of CAC between patients treated with different dialysis modality. They concluded that nocturnal hemodialysis and kidney transplantation are not associated with significantly less CAC progression compared to conventional hemodialysis during up to 3 years follow-up. I have several comments as follows.

1. This study takes much time and efforts to answer whether nocturnal HD or kidney transplantation decreases vascular calcifications. However, I think that its clinical significance and merit are limited because vascular calcification is not a good surrogate marker (Ref 1 and 2). A good surrogate marker should accurately reflect the clinical event they are supposed to surrogate. However, meta-analysis and randomized controlled studies did not show vascular calcifications reflecting the underlying risk for death in HD patients.

2. The authors mentioned “kidney transplantation is associated with less CAC progression compared to PD” in Discussion. However, I did not find any data in the article supporting this conclusion. Please shows the data or cite the references if available, otherwise please revise this sentence.

3. Because vascular calcification is just a surrogate marker for cardiovascular diseases or related mortality, the authors should explain why this study aims to investigate vascular calcification rather than major adverse cardiovascular events (MACE) or CV-related mortality as the primary outcome.

4. The article is not well organized, especially in Introduction and Discussion. I suggest rewrite some paragraphs to highlight the importance of your study. In addition, the repetition of several sentences is found in Discussion. Please revise them.

5. The authors should include baseline serum calcium, serum phosphorus, plasma PTH levels in as covariables in multivariate linear regression. These covariables for vascular calcification are more important compared to residual renal function or Vitamin K antagonists.

Ref 1: Con: Vascular calcification is a surrogate marker, but not the cause of ongoing vascular disease and it is not a treatment target in chronic kidney disease. Carmine Zoccali and Gerard London. Nephrol Dial Transplant (2015) 30: 352–357

Ref 2: Annual Progression of Coronary Calcification in Trials of Preventive Therapies

A Systematic Review. Peter A. McCullough and Kavitha M. Chinnaiyan. Arch Intern Med. 2009 Dec 14;169(22):2064-70.

Reviewer #2: The study enrolled 114 patients on dialysis and divided them into three groups: conventional hemodialysis, nocturnal hemodialysis, and kidney transplantation. The differences of changes of coronary artery calcification (CAC) with three-year period were evaluated using mixed-effect models for these groups. The author concluded that there is no significant difference of CAC progression in the group of kidney transplantation, and nocturnal hemodialysis compared to that in the group of conventional hemodialysis. The advantages of this study include well study design, transparent process in information collection, and correct statistical analysis. However, a main shortage of the study, insufficient sample size, may result in incorrect inference and misguide future research.

In addition, several comments were provided for improvement of this article.

1. I suggest the authors perform post-hoc power analysis. It can estimate type 2 error which represents the probability of false negative conclusion.

2. What are the indications for patients switching therapeutic model from conventional to nocturnal hemodialysis? If the indications are also related to CAC progression, the authors should probably consider to adjust when modeling.

3. Please compare patient characteristics between enrolled and not enrolled cases, which may possibly reflect the limits of generalization.

4. Line 89, Page 4: The sentence, “Patients left the study if they switched renal replacement therapy…” is ambiguous. Dose it also contain that patients switching therapeutic model from conventional to nocturnal hemodialysis? please modify it.

5. I also concern that the lack consistent patient numbers of peritoneal dialysis (n=31) with those (n=40) on previous published on Am J Nephrol 2018;48:369–377 because both number are from NOCTx study. Please explain it.

6. On Page 5, I am not sure whether the description, “dialysis duration defined from the first day to inclusion, minus the time with a functioning kidney transplantation” is correct or not. What does the word “inclusion” mean? Kidney translation should be after the time of inclusion, right?

7. Sum of not enrolled cases in the fig 1 is incorrect, please confirm it.

8. Is there any possible follow-up bias existing in your observation? The lost follow-up rate seems to be higher at 3rd year in the group of conventional hemodialysis than those in the other groups.

9. Does the discrepancies of cardiovascular disease / mortality risk between patients treated with conventional, nocturnal hemodialysis, or kidney transplant recipients probably contributed by the different CAC which reflects potential patient selection at baseline in this study? The author should discuss it.

6. PLOS authors have the option to publish the peer review history of their article (what does this mean?). If published, this will include your full peer review and any attached files.

Reviewer #1: Yes: CHIH-CHIA LIANG

Reviewer #2: No

---

## [Author Response · Author response to Decision Letter 0]

20 Mar 2020

Thank you for the opportunity to submit a revised version of our manuscript. Please find our responses to the comments below.

Editor:

It is important to include baseline serum calcium, serum phosphorus, plasma PTH levels as covariables in multivariate linear regression. 

We agree with the Editor and Reviewer #2 that it could be valuable to examine the associations of serum and plasma markers of mineral metabolism with CAC progression. However, we should not adjust for these variables when examining the association of renal replacement therapy with CAC progression. The reason is that in the case of renal replacement therapy, these variables are not confounders but mediators; i.e., kidney transplantation and nocturnal hemodialysis could have an effect on CAC progression by changing serum calcium, phosphorus, or plasma PTH[1]. Adjusting for these variables in multivariate regression models would lead to biased estimates instead of removing bias by confounding[2].

A characteristics comparison between enrolled and non-enrolled cases could be considered to reflect the generalizability. 

We agree with the Editor and Reviewer #2 that it would be valuable to compare characteristics of patients that were enrolled in the NOCTx study (n=181) with those who were not (n=148). However, as the patients that were not enrolled did not give informed consent for study participation, we could not collect any data on their characteristics. On the other hand, we do compare patient characteristics between enrolled patients that completed follow-up (n=114) and those that did not (n=36) in the Results section, page 11, lines 184-189:

“Patients that were excluded from the current analyses as they did not complete any follow-up exam (n=36) had similar characteristics compared to the study population: they were on average 52 ±13 years old, 23 (64%) were male, dialysis duration was median 36 (IQR 18–66) months, and 10 (28%) had diabetes mellitus (all P>0.05 versus study population). Their median CAC score at inclusion was 323 (IQR 1–1181) (P=0.18 versus study population).”

Also, we have now added a S1 Table comparing further characteristics between enrolled patients that completed follow-up and those that did not.

The lost follow-up is an important issue in this study that the author need to address this issue.

The Editor is right to point out that there was significant loss to follow-up in our study. Specifically, only 32 out of 57 enrolled patients on conventional hemodialysis completed the first follow-up visit. This could have led to survivor bias. However, only 2 of these 25 patients had died, whereas 4 had withdrawn consent, 7 were lost to follow-up, 7 had received a kidney transplant, and 5 had switched to nocturnal hemodialysis, leaving the study as a result. Therefore, we believe that the high rate of loss to follow-up in this group did not necessarily lead to a selection of healthier patients. We now address this issue in the Discussion section, page 18:

“There was significant loss to follow-up, especially among patients on conventional hemodialysis (25 out of 57 enrolled patients). This could have led to survivor bias, although many patients were also lost to follow-up due to kidney transplantation (7/25) or switched to nocturnal hemodialysis (5/25).”

Reviewer #1: 

This article entitled “Progression of coronary artery calcification in conventional hemodialysis, nocturnal hemodialysis, and kidney transplantation” by Jansz TT et al investigated progression of CAC between patients treated with different dialysis modality. They concluded that nocturnal hemodialysis and kidney transplantation are not associated with significantly less CAC progression compared to conventional hemodialysis during up to 3 years follow-up. I have several comments as follows.

1. This study takes much time and efforts to answer whether nocturnal HD or kidney transplantation decreases vascular calcifications. However, I think that its clinical significance and merit are limited because vascular calcification is not a good surrogate marker (Ref 1 and 2). A good surrogate marker should accurately reflect the clinical event they are supposed to surrogate. However, meta-analysis and randomized controlled studies did not show vascular calcifications reflecting the underlying risk for death in HD patients.

3. Because vascular calcification is just a surrogate marker for cardiovascular diseases or related mortality, the authors should explain why this study aims to investigate vascular calcification rather than major adverse cardiovascular events (MACE) or CV-related mortality as the primary outcome.

Thank you for pointing out this major controversy in the field of cardiovascular disease in end-stage renal disease. We fully agree with the reviewer that it is debatable whether vascular calcification is an adequate surrogate marker for cardiovascular disease. Even more, we believe our data support that vascular calcification may not be an adequate surrogate outcome. As we point out in the Discussion section, there is a discrepancy between our data not showing less vascular calcification progression among kidney transplant recipients, and the fact that kidney transplant recipients have significantly less cardiovascular morbidity and mortality. 

Unfortunately, our study was designed in 2008 before these discrepancies came to light. Therefore, our study was designed with CAC progression as primary outcome and was as such not powered for major adverse cardiovascular events or cardiovascular mortality as primary outcome. Nevertheless, we believe that future studies should use major adverse cardiovascular events or cardiovascular mortality instead of vascular calcification as an primary outcome, which we also state in the Discussion section, page 18:

“Previously, others have also pointed out important discrepancies between the effects of several drugs on vascular calcification and mortality[23]. For example, several trials have demonstrated that drugs such as cinacalcet or sevelamer may slow down progression of coronary artery calcification[24, 25] but lack benefit on mortality[25, 26]. These discrepancies suggest that vascular calcification might only be a secondary phenomenon to vascular damage and may not be harmful in itself. We therefore believe future studies should use major adverse cardiovascular events or cardiovascular mortality instead of vascular calcification as primary outcome.”

2. The authors mentioned “kidney transplantation is associated with less CAC progression compared to PD” in Discussion. However, I did not find any data in the article supporting this conclusion. Please shows the data or cite the references if available, otherwise please revise this sentence.

Please accept our apologies for this error. This sentence should have been deleted after a recent redrafting of this manuscript. We have now deleted this sentence accordingly.

4. The article is not well organized, especially in Introduction and Discussion. I suggest rewrite some paragraphs to highlight the importance of your study. In addition, the repetition of several sentences is found in Discussion. Please revise them.

As recommended by the reviewer, we revised several paragraphs in the Introduction and Discussion section. 

5. The authors should include baseline serum calcium, serum phosphorus, plasma PTH levels in as covariables in multivariate linear regression. These covariables for vascular calcification are more important compared to residual renal function or Vitamin K antagonists.

We agree with the Editor and Reviewer #2 that it could be valuable to examine the associations of serum and plasma markers of mineral metabolism with CAC progression. However, we should not adjust for these variables when examining the association of renal replacement therapy with CAC progression. The reason is that in the case of renal replacement therapy, these variables are not confounders but mediators; i.e., kidney transplantation and nocturnal hemodialysis could have an effect on CAC progression by changing serum calcium, phosphorus, or plasma PTH[1]. Adjusting for these variables in multivariate regression models would lead to biased estimates instead of removing bias by confounding[2].

Reviewer #2: The study enrolled 114 patients on dialysis and divided them into three groups: conventional hemodialysis, nocturnal hemodialysis, and kidney transplantation. The differences of changes of coronary artery calcification (CAC) with three-year period were evaluated using mixed-effect models for these groups. The author concluded that there is no significant difference of CAC progression in the group of kidney transplantation, and nocturnal hemodialysis compared to that in the group of conventional hemodialysis. The advantages of this study include well study design, transparent process in information collection, and correct statistical analysis. However, a main shortage of the study, insufficient sample size, may result in incorrect inference and misguide future research.

In addition, several comments were provided for improvement of this article.

1. I suggest the authors perform post-hoc power analysis. It can estimate type 2 error which represents the probability of false negative conclusion.

As suggested by the reviewer, we performed a post-hoc power analysis. We calculated the sample size needed to detect an effect of the magnitude suggested for kidney transplantation compared to conventional hemodialysis after a 1-year interval by our data, with 80% power and a significance level of 0.05. This yielded a sample size of 359 per group. We now address this issue in the Discussion section, page 18:

“We performed a post-hoc power analysis, revealing that over 350 patients per group would be needed to detect the difference currently observed between kidney transplant recipients and patients on conventional hemodialysis in a one-year interval with 80% power at a 0.05 significance level.”

2. What are the indications for patients switching therapeutic model from conventional to nocturnal hemodialysis? If the indications are also related to CAC progression, the authors should probably consider to adjust when modeling.

There are various reasons why patients wanted to switch to nocturnal hemodialysis from conventional hemodialysis. Important advantages of nocturnal hemodialysis include more free time during the day and less dietary restrictions. As a result, patients initiating nocturnal hemodialysis in general are often younger, healthier, more motivated, and more likely to adhere to treatment. Although we did not know the reasons of individual patients to switch to nocturnal hemodialysis, we did adjust for variables that may reflect overall health and thus could influence choice of renal replacement therapy as well as CAC progression (e.g. age, diabetes, and vitamin K antagonist use).

3. Please compare patient characteristics between enrolled and not enrolled cases, which may possibly reflect the limits of generalization.

We agree with the Editor and Reviewer #2 that it would be valuable to compare characteristics of patients that were enrolled in the NOCTx study (n=181) with those who were not (n=148). However, as the patients that were not enrolled did not give informed consent for study participation, we could not save any data on their characteristics. On the other hand, we do compare patient characteristics between enrolled patients that completed follow-up (n=114) and those that did not (n=36) in the Results section, page 11, lines 184-189:

“Patients that were excluded from the current analyses as they did not complete any follow-up exam (n=36) had similar characteristics compared to the study population: they were on average 52 ±13 years old, 23 (64%) were male, dialysis duration was median 36 (IQR 18–66) months, and 10 (28%) had diabetes mellitus (all P>0.05 versus study population). Their median CAC score at inclusion was 323 (IQR 1–1181) (P=0.18 versus study population).”

Also, we have now added a S1 Table comparing further characteristics between enrolled patients that completed follow-up and those that did not.

4. Line 89, Page 4: The sentence, “Patients left the study if they switched renal replacement therapy…” is ambiguous. Dose it also contain that patients switching therapeutic model from conventional to nocturnal hemodialysis? please modify it.

We apologize for the confusion arising from our wording. Patients in the nocturnal hemodialysis group switched to nocturnal hemodialysis shortly before enrollment. Indeed, if patients switched renal replacement therapy after enrollment (for example from conventional hemodialysis to nocturnal hemodialysis), they left the study. For clarity, we added “after enrollment” to the sentence: 

“Patients left the study if they switched renal replacement therapy after enrollment”

5. I also concern that the lack consistent patient numbers of peritoneal dialysis (n=31) with those (n=40) on previous published on Am J Nephrol 2018;48:369–377 because both number are from NOCTx study. Please explain it.

There is indeed a discrepancy in these numbers. The explanation is that a total of 4 patients were enrolled while on peritoneal dialysis (n=2) and conventional hemodialysis (n=2), but received a kidney transplantation >6 months after enrollment. These patients therefore had to leave the study, but requested re-enrollment. They were subsequently re-enrolled as kidney transplant recipients. For this paper, we used the data of those 4 patients after kidney transplantation, whereas in the paper Am J Nephrol 2018;48:369–377 we used data of these patients before kidney transplantation. Hence, there is a discrepancy of 2 patients in the peritoneal dialysis group (21 vs 23) and 2 patients in the conventional hemodialysis group (32 vs 34) between this paper and Am J Nephrol 2018;48:369–377. 

6. On Page 5, I am not sure whether the description, “dialysis duration defined from the first day to inclusion, minus the time with a functioning kidney transplantation” is correct or not. What does the word “inclusion” mean? Kidney translation should be after the time of inclusion, right?

We apologize for the confusion arising from our wording. Patients that were enrolled as kidney transplant recipients had received a kidney transplant shortly before enrollment. We added this to the Methods section, page 4, lines 88-89:

“NOCTx included [..] patients on dialysis who received a kidney transplant 2-3 months before enrollment.”

Furthermore, some patients had had kidney transplants longer before but were again on dialysis at time of enrollment. Their time with a kidney transplant was not counted towards dialysis duration. When mentioning inclusion in this context, we meant enrollment. For clarity, we replaced the word inclusion with enrollment throughout the manuscript. 

7. Sum of not enrolled cases in the fig 1 is incorrect, please confirm it.

Please accept our apologizes for this error. The number of patients with “other reasons” should be 57 instead of 56. We have amended the figure accordingly. 

8. Is there any possible follow-up bias existing in your observation? The lost follow-up rate seems to be higher at 3rd year in the group of conventional hemodialysis than those in the other groups.

Thank you for addressing this issue. Indeed, the rate of loss to follow-up was higher in the conventional hemodialysis group than in the nocturnal hemodialysis or kidney transplantation group. Notably, more than a third of the patients on conventional hemodialysis (34%) were lost to follow-up because they received a kidney transplant, which we mention in the Results section, page 11. We now also acknowledge this issue in the Discussion section, page 18:

“There was significant loss to follow-up, especially among patients on conventional hemodialysis”

9. Does the discrepancies of cardiovascular disease / mortality risk between patients treated with conventional, nocturnal hemodialysis, or kidney transplant recipients probably contributed by the different CAC which reflects potential patient selection at baseline in this study? The author should discuss it.

We thank the reviewer for addressing the issue of potential selection bias. Our study is observational and as such cannot fully eliminate potential selection bias. We attempted to minimize this by only enrolling patients on dialysis that were eligible for kidney transplantation and kidney transplant recipients that had been on dialysis. As a result, the patient groups were similar e.g. in terms of age and dialysis duration, although diabetes mellitus was somewhat more frequent in patients on conventional hemodialysis versus kidney transplant recipients (6/32 versus 3/48, P=0.22), which we adjusted for in our statistical analyses. We now discuss this issue in the limitations paragraph of the Discussion:

“Finally, our study was observational, which could potentially lead to selection bias. To minimize this, we only enrolled patients on dialysis who were transplantation-eligible and kidney transplant recipients that had been on dialysis before transplantation. The patient groups were comparable as a result”

References:

1. Rocco MV, Lockridge RS Jr, Beck GJ, Eggers PW, Gassman JJ, Greene T, Larive B, Chan CT, Chertow GM, Copland M, Hoy CD, Lindsay RM, Levin NW, Ornt DB, Pierratos A, Pipkin MF, Rajagopalan S, Stokes JB, Unruh ML, Star RA, Kliger AS; Frequent Hemodialysis Network (FHN) Trial Group, Kliger A, Eggers P, Briggs J, Hostetter T, Narva A, Star R, Augustine B, Mohr P, Beck G, Fu Z, Gassman J, Greene T, Daugirdas J, Hunsicker L, Larive B, Li M, Mackrell J, Wiggins K, Sherer S, Weiss B, Rajagopalan S, Sanz J, Dellagrottaglie S, Kariisa M, Tran T, West J, Unruh M, Keene R, Schlarb J, Chan C, McGrath-Chong M, Frome R, Higgins H, Ke S, Mandaci O, Owens C, Snell C, Eknoyan G, Appel L, Cheung A, Derse A, Kramer C, Geller N, Grimm R, Henderson L, Prichard S, Roecker E, Rocco M, Miller B, Riley J, Schuessler R, Lockridge R, Pipkin M, Peterson C, Hoy C, Fensterer A, Steigerwald D, Stokes J, Somers D, Hilkin A, Lilli K, Wallace W, Franzwa B, Waterman E, Chan C, McGrath-Chong M, Copland M, Levin A, Sioson L, Cabezon E, Kwan S, Roger D, Lindsay R, Suri R, Champagne J, Bullas R, Garg A, Mazzorato A, Spanner E, Rocco M, Burkart J, Moossavi S, Mauck V, Kaufman T, Pierratos A, Chan W, Regozo K, Kwok S. The effects of frequent nocturnal home hemodialysis: the Frequent Hemodialysis Network Nocturnal Trial. Kidney Int. 2011 Nov;80(10):1080-91.

2. Jager KJ, Zoccali C, MacLeod A, Dekker FW. Confounding: what is it and how to deal with it. Kidney Int 2008 Feb;73(3):256-60.

---

## [Decision Letter · Decision Letter 1]

22 Apr 2020

PONE-D-19-32023R1

Progression of coronary artery calcification in conventional hemodialysis, nocturnal hemodialysis, and kidney transplantation

PLOS ONE

Dear Dr. Thijs T Jansz,

Thank you for submitting your manuscript to PLOS ONE. After careful consideration, we feel that it has merit but does not fully meet PLOS ONE’s publication criteria as it currently stands. Therefore, we invite you to submit a revised version of the manuscript that addresses the points raised during the review process.

We would appreciate receiving your revised manuscript by Jun 06 2020 11:59PM. To enhance the reproducibility of your results, we recommend that if applicable you deposit your laboratory protocols in protocols.io, where a protocol can be assigned its own identifier (DOI) such that it can be cited independently in the future. For instructions see: http://journals.plos.org/plosone/s/submission-guidelines#loc-laboratory-protocols

We look forward to receiving your revised manuscript.

Kind regards,

Ping-Hsun Wu, M.D.

Academic Editor

PLOS ONE

Additional Editor Comments (if provided):

A new statistician was invited to review this study. Confounding factors and selection bias may still persist to compare therapeutic approaches. Please response the comments from reviewer 3.

Reviewers' comments:

Reviewer's Responses to Questions

**Comments to the Author**

1. If the authors have adequately addressed your comments raised in a previous round of review and you feel that this manuscript is now acceptable for publication, you may indicate that here to bypass the “Comments to the Author” section, enter your conflict of interest statement in the “Confidential to Editor” section, and submit your "Accept" recommendation.

Reviewer #1: (No Response)

Reviewer #2: All comments have been addressed

Reviewer #3: (No Response)

2. Is the manuscript technically sound, and do the data support the conclusions?

Reviewer #1: (No Response)

Reviewer #2: Yes

Reviewer #3: Partly

3. Has the statistical analysis been performed appropriately and rigorously? 

Reviewer #1: (No Response)

Reviewer #2: Yes

Reviewer #3: No

4. Have the authors made all data underlying the findings in their manuscript fully available?

Reviewer #1: (No Response)

Reviewer #2: Yes

Reviewer #3: Yes

5. Is the manuscript presented in an intelligible fashion and written in standard English?

Reviewer #1: (No Response)

Reviewer #2: Yes

Reviewer #3: Yes

6. Review Comments to the Author

Reviewer #1: (No Response)

Reviewer #2: The authors have satisfactorily responded to all my listed questions. Therefore, I have no further questions.

Reviewer #3: I come to this revised paper as a new reviewer.

I must respectfully disagree with reviewer 2 who talks of post hoc power. Power is an estimate used in planning. The data are what they are, and the actual power is either zero or 1 because the effect is either significant or not. I can wholeheartedly recommend the paper "The Abuse of Power" by Hoenig et al in The American Statistician.

The issue of whether there is absence of evidence or evidence of absence is based upon the confidence intervals. The issue in interpretation is whether the extremes of the CI represent something worth knowing about. If, for example, 0.96 is worth knowing about then one cannot rule out a clinically meaningful benefit and the conclusion must be that the study is inconclusive.

The paper as it stands doesn't adequately explain how people entered one or other group for comparison.What were the selection factors? Clearly there must have been some choice made here and groups cannot be entirely comparable. Not can those differences be measured exclusively in the variables collected here. Table 1 needs tests for confounding here, and the reason for dialysis vs transplantation in particular is needed.

Please explain the differences in missing data rates in Table 2 - there is potential for bias if these are outcome related.

7. PLOS authors have the option to publish the peer review history of their article (what does this mean?). If published, this will include your full peer review and any attached files.

Reviewer #1: Yes: CHIH-CHIA LIANG

Reviewer #2: Yes: Ming-Yen Lin

Reviewer #3: No

---

## [Author Response · Author response to Decision Letter 1]

5 Jun 2020

Thank you for the opportunity to submit a revised version of our manuscript. Please find our responses to the comments below.

Reviewer 3.

1. I must respectfully disagree with reviewer 2 who talks of post hoc power. Power is an estimate used in planning. The data are what they are, and the actual power is either zero or 1 because the effect is either significant or not. I can wholeheartedly recommend the paper "The Abuse of Power" by Hoenig et al in The American Statistician.

The issue of whether there is absence of evidence or evidence of absence is based upon the confidence intervals. The issue in interpretation is whether the extremes of the CI represent something worth knowing about. If, for example, 0.96 is worth knowing about then one cannot rule out a clinically meaningful benefit and the conclusion must be that the study is inconclusive.

Thank you for pointing this out. We have replaced the paragraph about post-hoc power in the Discussion with a paragraph in which we address that the confidence intervals include a possible clinically meaningful benefit:

“The 95% confidence intervals of the effect estimates for kidney transplantation include a more than two thirds lower CAC progression rate (adjusted minimum of the 95% confidence interval -0.96), which we consider clinically meaningful.”

2. The paper as it stands doesn't adequately explain how people entered one or other group for comparison. What were the selection factors? Clearly there must have been some choice made here and groups cannot be entirely comparable. Nor can those differences be measured exclusively in the variables collected here. Table 1 needs tests for confounding here, and the reason for dialysis vs transplantation in particular is needed.

The reviewer rightfully addresses the issue of how participants ended up in one treatment group or another, and whether there may be confounder bias beyond the variables presented in Table 1. 

In this study, there was no choice made for dialysis versus transplantation. As stated in the Methods and Discussion sections, all patients on dialysis were eligible for transplantation. Hence, the patients on dialysis had not received a kidney transplant, simply because they did not have the luck to receive one yet. Notably, 18 patients on dialysis did have the luck to receive a kidney transplant during study participation.

As for conventional hemodialysis versus nocturnal hemodialysis, there are various reasons why patients opted for nocturnal hemodialysis versus conventional hemodialysis. Important advantages of nocturnal hemodialysis include more free time during the day and less dietary restrictions. As a result, patients opting for nocturnal hemodialysis are often younger, healthier, more motivated, and more likely to adhere to treatment. Although we did not know the reasons of individual patients to opt for nocturnal hemodialysis (which we now acknowledge in the Limitations paragraph of the Discussion), we did adjust for variables that may reflect overall health and thus could influence choice of renal replacement therapy as well as CAC progression (e.g. age, diabetes, and vitamin K antagonist use). 

“For example, we did not know the reasons of individual patients to opt for nocturnal hemodialysis.”

Following the reviewer’s recommendation, we added significance tests to Table 1. Of note, there were no significant differences in demographics, medical history, and vitamin K antagonist use between treatment groups. There were significant differences in medication use and physical/laboratory parameters, expectedly due to the treatments received. However, we agree with the reviewer that there may be confounding beyond the variables measured in this study. Therefore, we added the following line to the Limitations paragraph of the Discussion:

“Although there may have been unmeasured confounders, patient groups on dialysis versus kidney transplant recipients were comparable in terms of the measured variables as a result.”

3. Please explain the differences in missing data rates in Table 2 - there is potential for bias if these are outcome related.

The reviewer is right to point out that there were differences in loss to follow-up in our study. Specifically, only 32 out of 57 enrolled patients on conventional hemodialysis completed the first follow-up visit. This could have led to survivor bias. However, only 2 of these 25 patients had died, whereas 4 had withdrawn consent, 7 were lost to follow-up, 7 had received a kidney transplant, and 5 had switched to nocturnal hemodialysis, leaving the study as a result. Therefore, we believe that the high rate of loss to follow-up in this group did not necessarily lead to a selection of healthier patients. We address this issue in the Discussion section, page 18:

“There was significant loss to follow-up, especially among patients on conventional hemodialysis (25 out of 57 enrolled patients). This could have led to survivor bias. Nevertheless, many patients were lost to follow-up due to kidney transplantation (7/25) or switch to nocturnal hemodialysis (5/25). Therefore, we do not believe that the high rate of loss to follow-up in this group led to a selection of healthier patients.”

---

## [Decision Letter · Decision Letter 2]

18 Sep 2020

PONE-D-19-32023R2

Progression of coronary artery calcification in conventional hemodialysis, nocturnal hemodialysis, and kidney transplantation

PLOS ONE

Dear Dr. Thijs T Jansz,

Thank you for submitting your manuscript to PLOS ONE. After careful consideration, we feel that it has merit but does not fully meet PLOS ONE’s publication criteria as it currently stands. Therefore, we invite you to submit a revised version of the manuscript that addresses the points raised during the review process.

We look forward to receiving your revised manuscript.

Kind regards,

Ping-Hsun Wu, M.D.

Academic Editor

PLOS ONE

Additional Editor Comments (if provided):

Most of the comments had been well response by the author. However, Reviewer 3 still have some comment on the patient selection in the different groups. Please clarify accordingly.

Reviewers' comments:

Reviewer's Responses to Questions

**Comments to the Author**

1. If the authors have adequately addressed your comments raised in a previous round of review and you feel that this manuscript is now acceptable for publication, you may indicate that here to bypass the “Comments to the Author” section, enter your conflict of interest statement in the “Confidential to Editor” section, and submit your "Accept" recommendation.

Reviewer #1: All comments have been addressed

Reviewer #2: All comments have been addressed

Reviewer #3: (No Response)

2. Is the manuscript technically sound, and do the data support the conclusions?

Reviewer #1: Yes

Reviewer #2: Yes

Reviewer #3: (No Response)

3. Has the statistical analysis been performed appropriately and rigorously? 

Reviewer #1: I Don't Know

Reviewer #2: Yes

Reviewer #3: (No Response)

4. Have the authors made all data underlying the findings in their manuscript fully available?

Reviewer #1: Yes

Reviewer #2: Yes

Reviewer #3: (No Response)

5. Is the manuscript presented in an intelligible fashion and written in standard English?

Reviewer #1: Yes

Reviewer #2: Yes

Reviewer #3: (No Response)

6. Review Comments to the Author

Reviewer #1: (No Response)

Reviewer #2: (No Response)

Reviewer #3: Thank you for your clarification of the design of the study. As I understand it patients will naturally change groups - what is unclear is why patients are excluded for compliance, as this leads to differently selected patients in the different groups, and why patients leave the study on switching therapy as opposed to changing group.

The analysis of this sort of study would appear to be the same as one would use for any other transplantation study. It is unclear why Mantel-Byar or equivalent analyses were not used for time to event outcomes and equivalent methodology for multiperiod crossover trials for the other measures. As it stands the different groups appear to enter the study post diagnosis, or time from starting dialysis - how is this allowed for in the analyses as the patients entering after a long period on dialysis are clearly more compliant and have disease that has not led to death (ie could be considered to be more indolent). As it stands the data aren't really interpretable since we are not looking at similar times.

7. PLOS authors have the option to publish the peer review history of their article (what does this mean?). If published, this will include your full peer review and any attached files.

Reviewer #1: No

Reviewer #2: **Yes: **Ming-Yen Lin

Reviewer #3: No

---

## [Author Response · Author response to Decision Letter 2]

5 Oct 2020

Thank you for the opportunity to revise our manuscript. Please find our responses to the queries raised by Reviewer #3 below.

Reviewer #3:

Thank you for your clarification of the design of the study. As I understand it patients will naturally change groups - what is unclear is why patients are excluded for compliance, as this leads to differently selected patients in the different groups.

We appreciate your concern as to why patients were excluded from the study if they were non-compliant, i.e. non-adherent to dialysis regimes. This was a deliberate choice when we designed the study, for the following. Often, patients initiating nocturnal hemodialysis are younger, healthier, more motivated, and more likely to adhere to treatment compared to the general dialysis population. We therefore sought to minimize these differences by selecting patients on conventional hemodialysis that were somewhat similar in this respect to patients on nocturnal hemodialysis, using treatment adherence as inclusion criterion. Without this criterion, patients on conventional hemodialysis could overall have been less compliant, which theoretically could have led to poorer phosphate control and hence more CAC progression in that group. We therefore believe that rather than introducing bias, this exclusion criterion contributed to reducing bias in our study.

To clarify this, we added the following explanation to the Discussion:

“On the other hand, we used treatment adherence as an inclusion criterion in order to enroll patients on conventional hemodialysis that were somewhat similar in this respect to patients on nocturnal hemodialysis, who are generally more likely to adhere to treatment”

[What is unclear is] why patients leave the study on switching therapy as opposed to changing group. The analysis of this sort of study would appear to be the same as one would use for any other transplantation study. It is unclear why Mantel-Byar or equivalent analyses were not used for time to event outcomes and equivalent methodology for multiperiod crossover trials for the other measures.

We agree with the reviewer that ideally patients would change group rather than leave the study upon switching therapy during study participation, especially in a study with survival as primary endpoint. However, our study did not investigate survival, but had CAC progression as primary endpoint. As a result, it was not feasible for patients to change groups during study participation for the following reasons. First, a switch in therapy would not coincide with the annual follow-up visits during which we measured CAC. Had we decided to allow patients to continue study participation after switching therapy, we would have been left with CAC progression measurements that had been influenced by multiple therapies, which would be difficult if not impossible to interpret. Second, patients would often switch to a hemodialysis therapy that did not match the hemodialysis categories defined in our study (3x4hrs/week or 4-6x8hrs/week). Many would switch to hemodialysis 3x8hrs/week, 4-6x2-4hrs/week, or anything in between. This also precluded continued participation after switching therapy.

For the above reasons, our study did not allow continued participation after switching therapy, and cannot be analyzed as a multiperiod crossover trial. We note that our study did not investigate time-to-event outcomes.

As it stands the different groups appear to enter the study post diagnosis, or time from starting dialysis - how is this allowed for in the analyses as the patients entering after a long period on dialysis are clearly more compliant and have disease that has not led to death (ie could be considered to be more indolent). As it stands the data aren't really interpretable since we are not looking at similar times.

Thank you for pointing out the issue of patients who have had a long period of dialysis (i.e. dialysis duration or vintage). Patients with a longer dialysis duration are a selected group: mortality has been shown to be the highest in the first 120 days after hemodialysis initiation[Robinson, 2014]. One could thus regard patients with a longer dialysis duration as the “survivors” of the dialysis population – this goes particularly for patients on nocturnal hemodialysis, who generally have had 1-2 years of dialysis before switching to this therapy. This is why we aimed to recruit patients on conventional hemodialysis and kidney transplant recipients who were somewhat similar to patients on nocturnal hemodialysis, using inclusion criteria such as treatment adherence, transplantation eligibility, and a period of dialysis for kidney transplant recipients. Notably, if we look at Table 1, we can appreciate that there were no significant differences in demographics or medical history between the groups, and that dialysis duration was actually quite similar in these three groups (median 27 months in conventional hemodialysis, 29 months in nocturnal hemodialysis, and 28 months in kidney transplant recipients). Thus, patients entered our study after a long period of dialysis, but this was similar for all three therapies. Moreover, we accounted for dialysis duration in our statistical analyses. We therefore do not believe that differences in dialysis duration would have biased our findings with regards to CAC progression.

We now state that the treatment groups were comparable in terms of dialysis duration in the Discussion:

“[..]We used treatment adherence as an inclusion criterion [and] only enrolled patients on dialysis who were transplantation-eligible [or] kidney transplant recipients that had been on dialysis before transplantation. Although there may have been unmeasured confounders, patient on dialysis versus kidney transplant recipients were [..] comparable in terms of baseline variables, notably age and dialysis duration”

References:

Robinson BM, Zhang J, Morgenstern H, et al. Worldwide, mortality risk is high soon after initiation of hemodialysis. Kidney Int. 2014;85(1):158-165. doi:10

---

## [Decision Letter · Decision Letter 3]

15 Dec 2020

Progression of coronary artery calcification in conventional hemodialysis, nocturnal hemodialysis, and kidney transplantation

PONE-D-19-32023R3

Dear Dr. Thijs T Jansz,

We’re pleased to inform you that your manuscript has been judged scientifically suitable for publication and will be formally accepted for publication once it meets all outstanding technical requirements.

Kind regards,

Ping-Hsun Wu, M.D.

Academic Editor

PLOS ONE

---

## [Editor Report · Acceptance letter]

18 Dec 2020

PONE-D-19-32023R3 

Progression of coronary artery calcification in conventional hemodialysis, nocturnal hemodialysis, and kidney transplantation 

Dear Dr. Jansz:

I'm pleased to inform you that your manuscript has been deemed suitable for publication in PLOS ONE. Congratulations! Your manuscript is now with our production department. 

Kind regards, 

on behalf of

Dr. Ping-Hsun Wu 

Academic Editor

PLOS ONE